# Survival of Porcine Reproductive and Respiratory Syndrome Virus (PRRSV) in the Environment

**DOI:** 10.3390/vetsci11010022

**Published:** 2024-01-05

**Authors:** Valeria Lugo Mesa, Angie Quinonez Munoz, Nader M. Sobhy, Cesar A. Corzo, Sagar M. Goyal

**Affiliations:** 1Veterinary Diagnostic Laboratory, Veterinary Population Medicine Department, College of Veterinary Medicine, University of Minnesota, St. Paul, MN 55108, USA; lugom003@umn.edu (V.L.M.); angieqm2509@gmail.com (A.Q.M.); nyaacoob@umn.edu (N.M.S.); corzo@umn.edu (C.A.C.); 2Facultad de Medicina Veterinaria y de Zootecnia, Universidad Nacional de Colombia, Bogotá 111321, Colombia

**Keywords:** PRRSV, virus survival, virus stability, environment, temperature

## Abstract

**Simple Summary:**

This review summarizes the available information on the ex-vivo survival of the porcine reproductive and respiratory syndrome virus (PRRSV), the cause of porcine reproductive and respiratory syndrome in pigs. We highlight the possible role of different fomites and environmental sources in indirect transmission of this virus to susceptible hosts. The number of studies on this topic is limited but fomites (porous, non-porous, and liquid), insects, people, and pork meat have been studied, mostly under experimental conditions.

**Abstract:**

Porcine reproductive and respiratory syndrome (PRRS) is one of the most economically important diseases of swine, with losses due to poor reproductive performance and high piglet and growing pig mortality. Transmission of porcine reproductive and respiratory syndrome virus (PRRSV) may occur by both direct and indirect routes; the latter includes exposure to PRRSV-contaminated fomites, aerosols, and arthropod vectors. This review has collected available data on the ex-vivo environmental stability and persistence of PRRSV in an effort to highlight important sources of the virus and to determine the role of environmental conditions on the stability of the virus, especially temperature. The ex-vivo settings include fomites (solid, porous, and liquid fomites), insects, people, and pork meat, as well as the role of environmental conditions on the stability of the virus, especially temperature.

## 1. Introduction

Porcine reproductive and respiratory syndrome (PRRS) is an important disease of pigs that causes high economic losses due to poor reproductive performance and mortality [1,2,3]. The causative agent of this disease is the porcine respiratory and reproductive syndrome virus (PRRSV). Around the mid-1980s, outbreaks of a severe disease were affecting swine herds in the Midwestern United States (US). Clinical signs included reproductive losses (e.g., abortion, stillbirths, mummies) in late gestational sows, a high number of weak live-born pigs, pneumonia, reduced growth performance, and mortality in growing pigs [4]. At almost the same time, Germany was also experiencing outbreaks of a similar disease. The nature of the causative agent was unknown at the time and, hence, the disease was named “mystery swine disease” in the US and “porcine epidemic abortion and respiratory syndrome” or “blue-ear pig disease” in Europe [2,5]. In 1991, a research group in Lelystad, the Netherlands, fulfilled Koch’s postulates with an isolated RNA virus. Soon after, a similar virus was isolated in the US (VR-2332), followed by Canada and other countries [5,6]. In 1992, at a meeting in Saint Paul, Minnesota, international researchers met to discuss the latest knowledge about this disease while officially naming it as PRRS [7].

## 2. The PRRSV

The PRRSV belongs to the family *Arteriviridae* in the order *Nidovirales*. It is a positive-sense, single-stranded RNA virus with an approximately 15 kb genome packed in a nucleocapsid protein. The virus is enveloped by a lipid bilayer with surface glycoproteins and membrane proteins. The genome has 11 open reading frames (ORFs), with the two largest being ORF 1a and 1b (75% of the viral genome), which encode two large non-structural polyproteins. ORFs 2–7 encode the seven viral structural proteins. The PRRSV is currently classified into two viral genotypes, PRRSV-1 and PRRSV-2. Both types occur worldwide, with PRRSV-1 being predominantly found in Europe and PRRSV-2 in North and South America. These two viruses share around 55–70% of nucleotide identity and are believed to have evolved separately from a distant common ancestor [5,8,9,10,11]. A commonly used classification system for PRRSV-2 is based on the restriction fragment length polymorphism (RFLP) of ORF5, a gene that encodes GP5, the major and most variable surface glycoprotein. This methodology was developed to differentiate modified live vaccines from wild-type viruses. Recently, another system based on the phylogenetic analysis of ORF5 was proposed in which type-2 viruses were classified into nine lineages. Most lineages are dominated by North American strains, while lineages 3 and 4 are primarily detected in Asia. Lineage 1, possibly of Canadian origin, is currently the most prevalent lineage in the US swine herd [12]. PRRSV diversity is rapidly increasing via point mutations and recombination, due to which several highly pathogenic PRRSV strains have emerged that cause acute disease outbreaks [5,13].

Clinical signs of PRRS are highly variable depending on viral variant, immune status and age of the host, co-infections, and stage of the disease. Pigs on some farms may be subclinically infected while others may show severe signs of reproductive and/or respiratory disease. In sow herds, PRRS is mainly manifested as reproductive failure (e.g., abortion, mummies, weak live-born piglets, pre-weaning mortality, and return to estrous), which may be accompanied by anorexia, fever, and lethargy. In growing pig farms, the virus causes respiratory disease (e.g., sneezing, coughing), leading to decreased feed intake and growth. Mortality is a consequence of viral infection due to severe pneumonia and co-infections [6,14].

The transmission of PRRSV may occur both by direct and indirect contact. Direct transmission of this virus can occur when susceptible pigs come in contact (e.g., nose to nose, natural breeding) with an infectious pig. The indirect route of transmission may include exposure to PRRSV-contaminated fomites, other environmental materials, aerosols, and arthropod vectors [6]. Therefore, given the economic impact of this disease and the efforts devoted to the prevention of introduction through biosecurity measures, this review aimed to summarize the available data on the ex-vivo environmental stability and persistence of PRRSV. A literature review was conducted in February 2023 using three bibliographic databases (i.e., PubMed, Science Direct, and Wiley) in which no limitations regarding publishing date were considered, which allowed for the inclusion of manuscripts as early as 1992 through early 2023. The search was carried out with the following keywords: porcine reproductive and respiratory syndrome virus, PRRSV survival, PRRSV inactivation, PRRSV stability, PRRSV in environment, and PRRSV fomites.

## 3. Persistence of PRRSV in the Environment

### 3.1. Tissues and Serum

The stability of PRRSV under common temperature and time conditions was investigated for optimal transmission conditions. Piglets were experimentally infected with PRRSV and euthanized 7 days post inoculation for the collection of tissues (e.g., right lung, spleen, and thymus) and serum. The virus was isolated in a cell culture from all samples collected at the time of necropsy. However, percent virus isolation after tissues were stored at 25 °C for 24, 48, and 72 h was 47%, 14%, and 7%, respectively. In contrast, 85% of the samples were positive at 4 °C and −20 °C after 72 h. All serum samples yielded viable virus except for those stored at 25 °C for 72 h [15].

### 3.2. Solid Fomites

The role of fomites in PRRSV transmission as well as its survival on different materials is summarized in Table 1. When evaluating the survival of PRRSV on different fomites at room temperature (25–27 °C), coupons of stainless steel, plastic, and boot rubber were contaminated with the virus. The virus was isolated only at 0 and 30 min post contamination with a drastic reduction in virus titer [16]. Similarly, pilot studies were conducted to evaluate virus viability at –2 °C and 20 °C on different materials in the presence and absence of snow. These materials were inoculated with PRRSV (MN-30100 strain) and then swabbed at different times. Viral RNA was detected by RT-PCR on plastic, metal, cardboard, and Styrofoam at both temperatures with and without snow for up to 4 h post inoculation. On other materials, viral RNA was detected for up to 12 h post inoculation at −2 °C from concrete and rubber (with and without snow) and from linoleum at 20 °C. Although virus isolation results varied depending on the material tested, the highest survival time was 4 h for most of them under cold temperature (−2 °C) conditions [17]. These materials were also tested at warmer temperatures and in the presence and absence of soil. Viral RNA was detected for up to 8 h post inoculation on all sampled surfaces (i.e., plastic, metal, cardboard, Styrofoam, rubber) in the absence of soil at both 10 °C and 20 °C. When covered with soil, viral RNA was detected up to 2 h on concrete, plastic, and rubber at 10 °C, as well as on plastic, rubber, and linoleum at 20 °C. Virus isolation was not possible from most of these samples past 2 h post inoculation, with the exception of soil-free plastic as virus was isolated for up to 1 h post inoculation at both 10 and 20 °C, as well as rubber at 20 °C and Styrofoam at 10 °C [18].

On stainless-steel coupons inoculated with PRRSV (FL-12 strain) incubated at 4 °C and 33–35% relative humidity, the virus remained viable for up to 24 h. Moreover, the difference in virus titer after 1 h and 24 h was not statistically significant [19]. For aluminum and cardboard coupons inoculated with PRRSV (MN-184), half-lives were reported to be 50 min on both surfaces at 30 °C, 16 min on aluminum and 20 min on cardboard at 40 °C, and 21 min on aluminum and 16 min on cardboard at 50 °C. The recent PRRSV 1-4-4 L1C variant survived longer, even at higher temperatures. Its half-lives were 5 h on cardboard and 2 h on aluminum at 30 °C, 4 h on aluminums and 47 min on cardboard at 40 °C, and less than 15 min on aluminum and 47 min on cardboard at 50 °C [20].

**Table 1 vetsci-11-00022-t001:** Survival of PRRSV on different solid surfaces/fomites.

Surface or Fomite	Strain	Temp (°C)	RH	Persistence	Reference
Stainless steel	Isolate 92 (10356; NVSL)	25–27 °C	NR	30 min VI	[16]
	FL-12	4 °C	33–35%	24 h VI	[19]
Plastic	Isolate 92 (10356; NVSL)	25–27 °C	NR	30 min VI	[16]
	MN-30100	−2 °C covered with snow		4 h RNA and VI	[17]
		−2 °C no snow		4 h RNA; 2 h VI	
		20 °C covered with snow		4 h RNA	
		20 °C no snow		4 h RNA; 2 h VI	
		10 °C covered with soil		2 h RNA; 1 h VI	[18]
		10 °C no soil		8 h RNA; 4 h VI	
		20 °C covered with soil		2 h RNA and VI	
		20 °C no soil		8 h RNA and VI	
Metal	MN-30100	−2 °C covered with snow		4 h RNA and VI	[17]
		−2 °C no snow		4 h RNA and VI *	
		20 °C covered with snow		4 h RNA; 2 h VI	
		20 °C no snow		4 h RNA; 0.5 h VI	
		10 °C covered with soil		1 h RNA	[18]
		10 °C no soil		8 h RNA; 2 h VI	
		20 °C covered with soil		1 h RNA; 2 h VI	
		20 °C no soil		8 h RNA; 2 h VI	
Rubber	MN-30100	−2 °C covered with snow		12 h RNA; 4 h VI *	[17]
		−2 °C no snow		12 h RNA; 2 h VI	
		10 °C covered with soil		2 h RNA; 1 h VI	[18]
		10 °C no soil		8 h RNA; 2 h VI	
		20 °C covered with soil		2 h RNA; 1 h VI	
		20 °C no soil		8 h RNA; 4 h VI	
	Isolate 92 (10356; NVSL)	25–27 °C		30 min VI	[16]
Styrofoam	MN-30100	−2 °C covered with snow		4 h RNA and VI	[17]
		−2 °C no snow		4 h RNA and VI	
		20 °C covered with snow		4 h RNA and VI	
		20 °C no snow		4 h RNA and VI	
		10 °C covered with soil		1 h VI	[18]
		10 °C no soil		8 h RNA; 4 h VI	
		20 °C covered with soil		1 h RNA and VI	
		20 °C no soil		8 h RNA; 2 h VI	
Cardboard	MN-30100	−2 °C covered with snow		4 h RNA and VI	[17]
		−2 °C no snow		4 h RNA and VI *	
		20 °C covered with snow		4 h RNA; 1 h VI	
		20 °C no snow		4 h RNA; 1 h VI	
		10 °C covered with soil		ND	[18]
		10 °C no soil		8 h RNA	
		20 °C covered with soil		1 h RNA	
		20 °C no soil		8 h RNA	
	MN-184	30 °C		50 min VI (HL)	[20]
		40 °C		20 min VI (HL)	
		50 °C		16 min VI (HL)	
	1-4-4 L1C	30 °C		5 h VI (HL)	
		40 °C		47 min VI (HL)	
		50 °C		47 min VI (HL)	
Concrete	MN-30100	−2 °C covered with snow		12 h RNA; 4 h VI *	[17]
		−2 °C no snow		12 h RNA; 4 h VI *	
		10 °C covered with soil		2 h RNA	[18]
		10 °C no soil		8 h RNA; 1 h VI	
Linoleum	MN-30100	20 °C covered with snow		12 h RNA; 4 h VI *	[17]
		20 °C no snow		12 h RNA; 2 h VI	
		20 °C covered with soil		2 h RNA; 1 h VI	[18]
		20 °C no soil		8 h RNA	
Aluminum	MN-184	30 °C		50 min VI (HL)	[20]
		40 °C		16 min VI (HL)	
		50 °C		21 min VI (HL)	
	1-4-4 L1C	30 °C		2 h VI (HL)	
		40 °C		4 h VI (HL)	
		50 °C		<15 min VI (HL)	

* VI results are not positive on all sampling points up to this point. RNA: Viral genetic material detected by PCR. VI: Viable virus detected in cell culture. SBA: Viable virus detected by swine bioassay. HL: Half-life. NR: Not reported. ND: Not detected.

### 3.3. Animal Feed

Animal feed has recently become a subject of research due to its possible role as a carrier of PRRSV. Several different animal feed ingredients have been studied, as shown in Table 2. Inoculated samples of alfalfa kept at room temperature (25–27 °C) were virus-isolation positive up to 30 min post inoculation [16]. Recently, a simulated transcontinental model of a shipment of animal feed ingredients was used to determine the survival of swine viruses in these ingredients. Duplicate samples of each feed ingredient (5 g/sample) were inoculated with 100 µL of a mixture of three viruses, PRRSV SD 1-7-4, porcine epidemic diarrhea virus (PEDV), and Senecavirus A (SVA). The environmental conditions were programmed into an environmental chamber, with mean temperatures ranging between 4 and 10 °C and mean relative humidity from 26% to 91%. Viable PRRSV was detected in conventional soybean meal and distillers’ dried grains with solubles (DDGS) after 37 days via swine bioassay but not through virus isolation in cell culture [21].

To further validate the hypothesis of feed ingredients acting as fomites, a demonstration project was carried out in which 30 g samples of feed ingredients (organic and conventional soya bean meal, lysine, choline, and vitamin A) were spiked with a 2 mL inoculum of the same three viruses. The samples were stored in a container inside a commercial truck while being transported across the US for 21 days. Results showed a high degradation of PRRSV viral RNA, as tested by qRT-PCR. However, the virus remained viable in the organic and conventional soybean meal as determined by swine bioassay [22].

Recently, a study was conducted to assess the viability of PRRSV (strain P129) in soybean meal (SBM), dried distillers’ grains with solubles (DDGS), and complete feed (CF) compared to viral media at different temperatures (i.e., 4 °C, 23 °C and 37 °C). In general, virus viability as assessed by virus isolation in MARC-145 cells was related to temperature, with the virus being isolated more frequently at lower temperatures. Regarding each matrix, the virus remained viable in SBM longer than in DDGS and CF regardless of the temperature [23].

In efforts to find possible ways to prevent virus dissemination through animal feed, the effect of extended storage of feed on virus survival was tested. Once again, 30 g samples of different feed ingredients were inoculated with the 2 mL mixture of the three previously mentioned viruses. These samples were stored for 30 days under two different environmental conditions: indoor storage under controlled environmental temperature and outdoor storage exposed to a Minnesota winter. Mean indoor environmental conditions were 20.1 °C and 35% RH, while outdoors these were −8.8 °C and 77% RH. Viral RNA from all viruses was detected by qRT-PCR in all ingredients under both indoor and outdoor storage conditions at the end of the storage period. Only samples kept under outdoor winter conditions contained viable viruses as tested by swine bioassays [24].

Higher volumes of feed (1-metric-tonne totes), spiked with 10 mL of PRRSV-contaminated ice cubes and stored in a temperature-controlled trailer for 30 days at 23.9 °C (mean RH of 62.4%), 15.5 °C (mean RH of 63.4%), and 10 °C (mean RH of 27.5%), yielded no viable virus by swine bioassay at 23.9 °C and 15.5 °C. However, at 10 °C, PRRSV RNA was detected from oral fluid samples collected from the pigs in the bioassay [25]. These totes of feed were also transported across different states in the US for 23 days inside a semi-trailer truck. The mean temperatures inside conventional and organic soybean meal feed totes were 9.4 °C and 7.9 °C, while mean RH was 66% and 21%, respectively. By the time the transport period ended, PRRSV RNA was detected by qRT-PCR in one out of two totes of organic and conventional soybean meal. Infection with PRRSV was confirmed in oral fluids collected from pigs inoculated with the positive feed samples. No PRRSV RNA was detected in conventional feed samples, and pigs naturally fed the said feed were not infected [26].

**Table 2 vetsci-11-00022-t002:** Survival of PRRSV in different animal feed ingredients.

Feed Ingredient	Strain	Temp (°C)	RH	Persistence	Reference
Alfalfa	Isolate 92 (10356; NVSL)	25–27 °C	NR	30 min VI	[16]
Straw					
Conventional soybean meal	1-7-4	4–10 °C	Range 26–91%	37 days SBA	[21]
		Mean 9.3 °C; Range −17 °C to 28 °C	Mean 35.7%; Range 10.4 to 75.3%	21 days RNA and SBA	[22]
		Mean 20.1 °C; Range 19.8 °C to 20.4 °C	Mean 35%; Range 34–37%	30 days RNA	[24]
		Mean −8.8 °C; Range −4 °C to 14.7 °C	Mean 77%; Range 62–88%	30 days RNA and SBA	
		Mean 9.4 °C; Range 3.2 °C to 17 °C	Mean 66%; Range 38% to 66%	23 days RNA and SBA	[26]
	1-4-4 L1C	23.9 °C	Mean 62.4%	ND	[25]
		15.5 °C	Mean 63.4%	ND	
		10 °C	Mean 27.5%	30 days SBA	
Organic soybean meal	1-7-4	Mean 9.3 °C; Range −17 °C to 28 °C	Mean 35.7%; Range 10.4 to 75.3%	21 days RNA and SBA	[22]
		Mean 20.1 °C; Range 19.8 °C to 20.4 °C	Mean 35%; Range 34 to 37%	30 days RNA	[24]
		Mean −8.8 °C; Range −4 °C to 14.7 °C	Mean 77%; Range 62 to 88%	30 days RNA and SBA	
		Mean 9.4 °C; Range 3.2 °C to 17 °C	Mean 66%; Range 38% to 66%	23 days RNA and SBA	[26]
Vitamin A	1-7-4	Mean 9.3 °C; Range −17 °C to 28 °C	Mean 35.7%; Range 10.4 to 75.3%	21 days RNA and SBA	[22]
		Mean 20.1 °C; Range 19.8 °C to 20.4 °C	Mean 35%; Range 34 to 37%	30 days RNA	[24]
		Mean −8.8 °C; Range −4 °C to 14.7 °C	Mean 77%; Range 62 to 88%	30 days RNA and SBA	
Choline chloride	1-7-4	Mean 20.1 °C; Range 19.8 °C to 20.4 °C	Mean 35%; Range 34 to 37%	30 days RNA	
		Mean −8.8 °C; Range −4 °C to 14.7 °C	Mean 77%; Range 62 to 88%	30 days RNA and SBA	
Lysine	1-7-4	Mean 20.1 °C; Range 19.8 °C to 20.4 °C	Mean 35%; Range 34 to 37%	30 days RNA	
		Mean −8.8 °C; Range −4 °C to 14.7 °C	Mean 77%; Range 62 to 88%	30 days RNA and SBA	
Complete feed	1-7-4	Mean 9.4 °C; Range 3.2 °C to 17 °C	Mean 66%; Range 38% to 66%	ND	[26]

RNA: Viral genetic material detected by PCR. VI: Viable virus detected in cell culture. SBA: Viable virus detected by swine bioassay. HL: Half-life. NR: Not reported. ND: Not detected.

### 3.4. Persistence of PRRSV in Fecal Slurry and Other Liquids

At room temperature (25–27 °C), the PRRSV inoculated in city and well water was found to be viable for 11 and 9 days, respectively. From saliva, urine, and fecal slurry, the virus was isolated only at 0 and 30 min post inoculation [16]. Other pig slurry samples inoculated with PRRSV VR-2332 were virus-isolation positive for up to 14 days when stored at 4 °C, up to 1 day at 25 °C, and 6 h at 37 °C [27]. Using a similar methodology, lagoon effluents from a nursery farm were inoculated with PRRSV MN-30100 and stored at different temperatures. At 4 °C, samples were RT-PCR positive for the whole 12-day duration of the study, while viable virus was detected by virus isolation for up to 8 days. At 20 °C, the samples were also RT-PCR positive throughout the 12-day period but no virus was isolated; however, samples were positive through swine bioassays for up to 3 days [28].

More recently, the half-life of PRRSV in pig manure at different temperatures was compared with a PRRSV-inoculated minimum essential medium (MEM), which was used as a control. The mean half-life of PRRSV at 4 °C was 112.6 and 120.5 h in manure and MEM, respectively. At 20 °C, the half-lives were 14.6 and 24.5 h; at 40 °C, 14.6 and 24.5 h; at 60 °C, 2.9 and 8.5 min; and at 80 °C, 0.36 and 0.59 min, respectively. Thus, an exponential decrease in PRRSV infectivity occurred as the temperature increased. However, the virus was more stable in MEM than in manure [29] (see Table 3).

### 3.5. Persistence of PRRSV in Aerosols

The detection and transmission of PRRSV in air was previously reviewed [33]. This review highlighted the challenges faced in detecting airborne PRRSV, which are mainly due to the differences between experimental and field settings. Experimental, semi/experimental, and field studies yielded conflicting results. The differences could have been due to the air sampling methods, underlying immunity of herds in field settings, environmental conditions, herd size and dynamics, and differences among virus strains in terms of their capability of being shed nasally, becoming airborne, and remaining infectious. The authors stated that there was indeed a possibility of airborne transmission of PRRSV, but the frequency and conditions under which this may happen have not been properly elucidated.

A few studies have been performed on the effect of environmental conditions on the stability of PRRSV in aerosols. The PRRSV VR-2332 was aerosolized under controlled conditions of temperature and relative humidity followed by the collection of air samples repeatedly over time. Results indicated that the virus had the least stability at 41 °C and 73% humidity, and the most stability at 5 °C and 17% relative humidity. The half-life of the virus was inversely proportional to temperature and relative humidity, with temperature having a higher effect on virus inactivation than relative humidity. Viral RNA quantities, on the other hand, remained stable under the above conditions [34].

While researching the airborne transmission of PRRSV, a building housing virus-positive pigs was located 120 m away from two recipient buildings, one filtered and one non-filtered. The odds of detecting PRRSV-positive bioaerosols in the non-filtered recipient building were higher when the predominant wind direction was towards the recipient building, barometric pressure was high, and the lowest relative humidity measurement of the day was present [35]. Wind velocity (m/s), wind gust (m/s), and sunlight intensity (watts/m^2^) also had a statistically significant impact on the airborne transport of the virus [36].

### 3.6. Persistence of PRRSV in/on Insects

Studies on the survival of PRRSV in insects have been mainly focused on evaluating the role of flies and mosquitoes as possible vectors (see Table 4) since these insects are commonly found in farms during the summer months. In most of these studies, insects were allowed to feed on PRRSV-positive pigs under controlled experimental conditions. In an initial experiment, only 1 of 22 pools of mosquitoes (30 *Aedes* spp. per pool) had detectable PRRSV RNA, which was 100% homologous to the virus isolated from the pigs the mosquitoes fed on. This virus was also viable as tested by swine bioassay [37]. Subsequently, the transmission of PRRSV from infected to naive pigs through mosquitoes (*Aedes vexans*) was investigated. Mosquitoes contained in a vial were allowed to feed on a PRRS-viremic pig until repletion and then on a recipient naive pig. The latter became PRRSV positive, with a 100% homologous virus. One of the pooled mosquito homogenates yielded positive RT-PCR while the other two were positive only by swine bioassay [38]. To determine how long the PRRSV could remain viable in these insects, 100 mosquitoes were kept at 27 °C for sampling at different time points. Only gut homogenates were positive for RT-PCR and swine bioassays at 0 and 6 h post feeding on a PRRSV-viremic pig. When these mosquitoes were kept for a longer period of time (i.e., 7 to 14 days), they were not able to transmit PRRSV to susceptible pigs. In addition, PRRSV RNA was not detected by RT-PCR. Thus, although PRRSV could survive in the intestinal tract of mosquitoes for up to 6 h, the virus did not replicate or disseminate within the mosquitoes after 7 to 14 days [39].

The possible role of another insect (houseflies, *Musca domestica Linnaeus*) as mechanical vectors of PRRSV was initially evaluated. A total of 100 houseflies per day were allowed to feed on a PRRSV-viremic pig followed by transference to feed on a recipient pig. In all three replicates, both donor and recipient pigs were PRRSV positive with 100% nucleic acid homology of the viruses. Although whole fly homogenates had detectable PRRSV RNA by RT-PCR, no viable virus could be isolated. Thus, houseflies had the potential to transmit PRRSV to susceptible pigs. The viability of the PRRSV was then evaluated at 27 °C in 210 flies that fed on a viremic pig. Pooled samples were collected at different time points and processed as whole fly homogenates. Subsets collected immediately and 6 h post feeding were positive by both RT-PCR and swine bioassay [40]. The survival time of the virus was then compared between processing methods for the flies. For whole fly homogenates, PRRSV RNA was detected at all sampling times (i.e., 0, 6, 12, 24 h post feeding), while swine bioassay was positive for samples collected up to 12 h post feeding. For exterior surface washes, PRRSV RNA was detected up to 12 h post feeding, but only the sample collected immediately after feeding was positive by swine bioassay. Gut homogenates were positive by RT-PCR and viable virus was detected by swine bioassay at 0 and 12 h post feeding [41]. By processing flies individually and not as pools, viral RNA was detected in 12 out of 13 gut homogenates and virus was isolated from a total of 7 homogenates. Individual flies were found to carry sufficient amounts of PRRSV to infect a susceptible pig after feeding on an infected pig [42].

To spatially analyze the survival of PRRSV, flies were collected from jug traps set at different distances from a facility that housed experimentally infected pigs. Viral RNA was detected in fly samples collected up to 2.3 km away from the facility, with 99.7–100% homology to the index virus. Viable virus was detected by swine bioassay in flies collected 0.4 km, 0.8 km, and 1.3 km away from the facility housing the pigs. However, the ability of these flies to transmit the virus to susceptible pigs was not assessed [43].

Temperature has also been found to have a considerable effect on the survival of PRRSV in flies. Under laboratory conditions, groups of flies that had fed on a viremic pig were placed at different temperatures for sampling over time. By both qRT-PCR and viral isolation, the processed pools of whole fly homogenates were positive for up to 48 h at 15 °C, 14–22 h at 20 °C, 12–18 h at 25 °C, and 12 h at 30 °C. When these houseflies were placed in containers under field conditions, where temperatures ranged from 9 °C to 22.5 °C and relative humidity from 32% to 99%, results showed that at 48 h, 30% to 40% of the flies still had detectable viral RNA. Randomly selected PCR-positive samples were also tested by swine bioassay, with positive results in 6 out of 13 samples. This indicated survival of infectious PRRSV in houseflies up to 48 hours at warm temperatures [44].

Another species of flies, stable flies (*Stomoxys calcitrans*), was experimentally exposed to PRRSV by feeding on viremic pigs. The virus was isolated from pooled samples of these flies for up to 24 h after feeding. However, these flies were not able to infect naive pigs. In a subsequent experiment, the flies were allowed to have a partial blood meal and then fed on naive piglets. Viral RNA was detected in fly pools for up to 12 h after the partial blood meal. Once again, none of the naive piglets became infected [45]. Stable flies that fed on blood spiked with live and inactivated PRRSV and thereafter kept at room temperature (20–25 °C) showed detection of viral RNA in pooled gut homogenates up to 48 h post feeding and the virus was isolated up to 24 h post feeding from the live active virus. For flies that fed on the inactivated virus, only viral RNA was able to be detected up to 24 h post feeding [46]. As part of a recent pilot study, stable flies from 20 pig farms in Austria were tested. Some of the farms were positive for PRRSV while others were negative. The mouth and abdominal parts of all flies were negative by qRT-PCR [47]. These results suggest that stable flies are a minor, non-common source of PRRSV transmission.

**Table 4 vetsci-11-00022-t004:** Survival of PRRSV in/on insects.

Insect	Strain	Temp (°C)	Sample	Persistence	Reference
Mosquitoes (*Aedes vexans*)	MN-30100	27 °C	Gut homogenates	6 h RNA and SBA	[39]
Houseflies (*Musca domestica Linnaeus*)			Whole fly homogenates	6 h RNA and SBA	[40]
				24 h RNA; 12 h SBA; ND VI	[41]
			Gut homogenates	12 h RNA and SBA; ND VI	
			External fly washes	12 h RNA; SBA only immediately post feeding; ND VI.	
		Max 22 °C to 37 °C, Min 8 °C to 19 °C	Whole fly homogenates	RNA detected in flies up to 2.3 km away from source	[43]
		15 °C		48 h RNA and VI	[44]
		20 °C		14–22 h RNA and VI	
		25 °C		12–18 h RNA and VI	
		30 °C		12 h RNA and VI	
		9 °C to 22.5 °C		>48 h (3.2 h HL) RNA and SBA	
Stable fly (*Stomoxys calcitrans*)	NC Powell/RespPRRS vaccine virus	18 °C	Body homogenates	24 h VI	[45]
				12 h RNA (after a partial blood meal)	
	VR-2332/Vaccine modified live virus	20–25 °C	Gut homogenates	48 h RNA (flies fed live active virus)	[46]
				24 h RNA (flies fed inactivated virus)	
				24 h VI (flies fed live active virus)	

RNA: Viral genetic material detected by PCR. VI: Viable virus detected in cell culture. SBA: Viable virus detected by swine bioassay. HL: Half-life. NR: Not reported. ND: Not detected.

### 3.7. Persistence of PRRSV on People

The role of people within the swine industry as a possible mechanical vector has also been considered (see Table 5). Ten individuals were allowed to have direct physical contact with PRRSV-infectious pigs for 1 continuous hour and then proceed to interact with PRRSV-susceptible pigs. Samples of nasal secretions, fingernail rinses, and saliva were collected from these people before and at different time points after exposure. Viral RNA was detected in saliva and fingernail rinse samples from two people directly after exposure. Even after showering, PRRSV RNA was detected in a fingernail rinse sample from one person at 5 h post exposure and a nasal swab sample from another person at 48 h post exposure. None of the naive pigs exposed to these individuals were infected. Therefore, after direct contact with infected pigs, humans were able to harbor the virus for up to 48 h but could not transmit it to susceptible pigs [48].

Similarly, individuals who had been exposed to PRRSV-viremic pigs were put in contact with susceptible pigs without allowing them to change contaminated boots and coveralls or to wash their hands to determine if they could transmit the virus. These naive pigs became infected, while those that were exposed to the individuals that followed biosecurity measures such as changing boots and coveralls, showering, washing hands, and having down time did not [49]. Under field conditions, it was also found that in the absence of sanitation measures, PRRSV could be transmitted by personnel and their clothes/boots. Three individuals would go to a PRRSV-negative pig facility (facility A) followed by visit to a facility housing infected pigs (facility B) and finally to another PRRSV-negative facility (facility C). They did not shower or change clothes between facilities B and C. Pigs in facility C eventually became infected with PRRSV. Also, viral RNA was detected in a majority of hand, boot, and coverall swabs collected in this facility. This virus was >99% homologous to the one that the pigs in facility B were infected with, confirming transmission of the virus between the two facilities [35].

To investigate the transmission of PRRSV through contaminated hands of personnel, meat juice collected from experimentally infected pigs (PRRSV MN-184) was applied onto the palms of individuals. Immediately after being contaminated with meat juice, a PRRSV-naive pig was handled with a contaminated hand. The pigs that came in contact with contaminated hands at 0 and 30 min after contamination were confirmed to be infected 7 days after contact [50].

**Table 5 vetsci-11-00022-t005:** Survival of PRRSV on people.

	Strain	Temp (°C)	Sample	Persistence	Reference
People	P-129	NR	Nasal secretions	48 h RNA	[48]
			Fingernail rinses	5 h RNA	
	MN-184	NR	Contact with naive pigs after hand contamination	30 min SBA	[50]

RNA: Viral genetic material detected by PCR. VI: Viable virus detected in cell culture. SBA: Viable virus detected by swine bioassay. HL: Half-life. NR: Not reported. ND: Not detected.

### 3.8. Persistence of PRRSV in Pig Meat

In packaged pig meat samples obtained from processing plants, no detectable amounts of PRRSV RNA were found nor was the virus isolated, indicating unlikely transmission of the virus through pig meat [51]. Out of 1027 samples collected from slaughterhouses in Canada, only 19 (1.9%) were RT-PCR positive for PRRSV. Moreover, only 1 out of these 19 positive samples yielded viable virus by isolation in cell culture. However, 6 out of 10 pigs fed some of these positive samples for two consecutive days became infected with PRRSV. The authors concluded that low amounts of PRRSV might be present in a small percentage of meat samples from pig slaughterhouses but that even this low dose could infect naive animals [52]. This low percentage of positive samples was also detected within 1500 fresh bottom meat samples collected from a ham boning plant in Canada; only 0.73% were RT-PCR positive [53].

Storage conditions of meat over time are another factor to take into account. Muscle tissue obtained from pigs 11 days after being experimentally infected was frozen at −23 °C for 10 days and then thawed for 24 h at 4 °C. Viral RNA was detected in most of the samples; however, viral titers decreased considerably. The virus was isolated from 3 out of 12 muscle samples of pigs infected with the Lelystad strain and 1 out of 12 samples from those infected with a North American strain (SDSU #73) [54]. Similarly, muscle tissue samples obtained 7 days after experimental infection with PRRSV (MN-184) were stored at −20 °C for 1 month. Thereafter, the samples were thawed at 4 °C for 7 days. Viral RNA was detected in thawed samples for up to 8 days by RT-PCR, and there was no significant decrease in viral RNA concentration during this time. The virus was found to have remained viable by swine bioassays [50].

Storage of pig meat samples at −24 °C and sampled weekly for up to 15 weeks, showed that only four of nine had a detectable viral load by qRT-PCR, which was relatively low and close to the limit of detection. However, swine bioassays were positive for PRRSV in five out of nine pigs after the end of the storage period. Although transmission of PRRSV from orally fed infected meat was observed, the overall probability of these samples carrying a high load of infectious virus was concluded to be low [53].

When spiked with 10^6^ TCID_50_/mL of PRRSV, the virus was detected in fresh sausage samples for up to 15 days at 4 °C and for up to 30 days at −20 °C. In bacon, the virus was detected only at time 0 (right after the preparation of the sample). In ham and acidified sausage, no virus was detected at any time point. Given the heat and/or chemical treatments that bacon, ham, and acidified sausage go through, there is a low likelihood of PRRSV transmission through these products. In contrast, fresh sausage was found to be a plausible vehicle for virus transmission [55].

The survival of different concentrations of PRRSV in pig meat samples has also been studied. Samples of fresh lean pork inoculated with different concentrations (i.e., 10^3^, 10^4^, 10^5^ TCID_50_/mL) of PRRSV VR-2332 were stored at 4 °C, 25 °C, and −20 °C. The virus was detected by virus isolation in samples stored at 25 °C for up to 48 h for all three virus concentrations, but with a sharp decline in the first 24 h. At 4 °C, viable virus was detected in samples inoculated with 10^5^ and 10^4^ TCID_50_/mL for 6 days and for up to 3 days in samples inoculated with 10^3^ TCID_50_/mL. At −20 °C, PRRSV was detected for up to 60 days in samples inoculated with 10^5^ and 10^4^ TCID_50_/mL, but only for 7 days when inoculated with the lowest virus concentration. These results indicate that the rate of PRRSV inactivation in fresh pork was dependent on the storage temperature and initial amount of virus present [56]. These survival studies on pig meat are summarized in Table 6.

### 3.9. Efect of Temperature on the Persistence of PRRSV

As mentioned before, temperature is an important factor on the survival of PRRSV in different environments/fomites. When suspended in MEM, PRRSV VR-2332 had a 50% reduction in titer after 12 h of incubation at 37 °C, as determined by microtitration assay in cell culture. Complete viral inactivation was observed after 48 h at 37 °C and after 45 min at 56 °C. However, the virus titer was stable for 1 and 4 months at 4 °C and −70 °C, respectively [30].

For four different PRRSV-2 isolates (VR-2332, JA-142, MN-184, and Ingelvac^®^ vaccine), RNA concentrations remained stable as determined by RT-PCR when these were stored at 4 °C, 10 °C, 20 °C, and 30 °C. However, virus infectivity, as calculated by microtitration assays, was not correlated to the RT-PCR results. The viral infectivity results were used to calculate half-lives, with the following results: 155.5 h at 4 °C, 84.8 h at 10 °C, 27.4 h at 20 °C, and 1.6 h at 30 °C. Although different temperatures had a statistically significant effect on virus half-life, various viral isolates behaved similarly at a given temperature [31].

The survival of 10 different PRRSV isolates (i.e., MN-184, 1-4-4 MN L1C, 1-4-4 SD L1C, Lelystad, VR-2332, 1-4-2, 1-26-2, ATP Vaccine, 2-5-2, 1-7-4) was studied at 4 °C, 25 °C, and 37 °C. All isolates survived for at least 35 days at 4 °C. At 25 °C, half the isolates survived no longer than 1 day, while VR-2332, Lelystad, 1-4-4 SD and MN, and MN-184 survived for 3 to 7 days. At 37 °C, only the Lelystad, 1-4-4 SD and MN, and MN-184 isolates survived for 3 days. The remaining isolates survived for no longer than 1 day. These results highlight the longer survival of the recent PRRSV 1-4-4 L1C at the three temperatures evaluated [32].

## 4. Discussion

Studies evaluating the persistence of PRRSV in the environment are limited and mostly experimental. However, there is an agreement that the virus survives the longest at colder temperatures. Solid non-porous fomites such as plastic and rubber may help the virus maintain its viability for a longer time. The fact that these materials are commonly found on supplies entering farms or are worn by workers highlights the importance of their proper cleaning and disinfection. These items may serve as a source of PRRSV, as it has been detected in these milieus even at warmer temperatures. Contaminated hands of workers themselves may also help transmit the virus. Other materials such as Styrofoam, cardboard, linoleum, concrete, and stainless steel have also been reported to harbor PRRSV for different amounts of time. Therefore, biosecurity measures incorporating the use of clean materials and work wear, as well as hand-washing, is an important strategy to reduce indirect transmission.

Regarding porous fomites, soybean-based feed showed a higher possibility of virus survival even after long periods of transport (37 days). This suggests a need for the re-evaluation of feed storage/transport conditions. Liquid fomites have not been widely researched as sources of PRRSV, although infectious virus has been isolated from city and well water as well as swine slurry. These findings suggest the possibility of waterborne/liquid transmission of PRRSV, especially in settings where water is not treated/chlorinated; however, this requires further evaluation.

Under laboratory conditions, insects (i.e., mosquitoes and flies) have been shown to be a potential source of PRRSV transmission to uninfected pigs. The virus has been isolated from fly samples for up to 48 h after a blood meal. Although this may be of potential concern, especially during the summer months, this indirect route of transmission is not of major concern compared to other routes and its true impact on PRRS incidence has not been evaluated.

While PRRSV viral RNA and infectious virus have been detected in pig meat products, the prevalence of positive samples is very low at processing plants. When experimentally inoculated in pig meat samples and kept at low temperature (4 °C or below), the virus may indeed persist for long periods of time. However, even though infected meat fed to pigs and infected human hands (from meat drippings) have been shown to be a possible source of transmission to susceptible pigs, the likelihood of this event happening seems to be low and requires further epidemiological assessments.

The virus remains viable for longer periods of time at lower temperatures, which automatically suggests that exposing fomites or surfaces at high temperatures for long periods of time will contribute to the inactivation and reduce the probability of virus transmission. Pig farms have either implemented or are considering using heat sources in enclosed areas to expose incoming farm supplies to inactivate virus that may have contaminated a given surface. However, the interaction between temperature and disinfectants needs to be further investigated in order to provide swine practitioners and producers with more tools to prevent the introduction of the virus into their farms.

Assessing the viability of PRRSV is currently being conducted either through cell culture or swine bioassay. While these two approaches provide important information regarding the virus, these can generate some degree of doubt given the sensitivity of each of these procedures. Regarding cell culture, it is well documented that some variants can be challenging to grow in contemporary cell cultures given poor virus adaptability, which can yield false-negative results. On the other hand, swine bioassay is an important tool but continues to remain costly and the level of infectiousness of each variant may be unknown or insufficient when exposing the pigs to these RT-PCR positive samples. Therefore, tools assessing the viability of the pathogen that do not depend on cells or a swine bioassay model will significantly contribute to the understanding of virus viability under different conditions.

## 5. Conclusions

In conclusion, the literature has important information of virus survival under various conditions. However, there is no uniformity in the methods of study, quality of the laboratory, test used to demonstrate virus viability, and virus strain being studied. A comprehensive study taking these factors into account is indicated.

## Figures and Tables

**Table 3 vetsci-11-00022-t003:** Survival of PRRSV in swine slurry and other liquids.

Liquid	Strain	Temp (°C)	Persistence	Reference
City water	Isolate 92 (10356; NVSL)	25–27 °C	11 days VI	[16]
Well water			9 days VI	
Swine saliva			30 min VI	
Swine urine			30 min VI	
Swine slurry			30 min VI	
	VR-2332	4 °C	14 days VI	[27]
		25 °C	1 day VI	
		37 °C	6 h VI	
	MN-30100	4 °C	12 days RNA; 8 days VI	[28]
		20 °C	12 days RNA; ND VI	
Pig manure	1-18-2 (L1, SL5)	4 °C	112.6 h VI (HL)	[29]
		20 °C	14.6 h VI (HL)	
		40 °C	1.6 h VI (HL)	
		60 °C	2.9 min VI (HL)	
		80 °C	0.36 min VI (HL)	
MEM	VR-2332	4 °C	>4 months VI	[30]
		37 °C	48 h VI (12 h HL)	
		56 °C	45 min VI	
		−70 °C	>4 months VI	
	VR-2332/JA-142/MN-184/ Ingelvac^®^ ATP vaccine virus	4 °C	155.5 h HL (VI)	[31]
		10 °C	84.9 h HL (VI)	
		20 °C	27.4 h HL (VI)	
		30 °C	1.6 h HL (VI)	
	MN-184/1-4-4 MN L1C/1-4-4 SD L1C/Lelystad/VR-2332/1-4-2/1-26-2/ Ingelvac^®^ ATP vaccine virus/2-5-2/1-7-4	4 °C	>35 days VI	[32]
	MN-184/1-4-4 MN L1C	25 °C	7 days VI	
	1-4-4 SD L1C/Lelystad/VR-2332		3 days VI	
	1-4-2/1-26-2/ Ingelvac^®^ ATP vaccine virus/2-5-2/1-7-4		1 day VI	
	MN-184/1-4-4 MN L1C/1-4-4 SD L1C/Lelystad	37 °C	3 days VI	
	VR-2332/1-4-2/1-26-2/ Ingelvac^®^ ATP vaccine virus/2-5-2/1-7-4		1 day VI	
	1-18-2 (L1, SL5)	4 °C	120.5 h VI (HL)	[29]
		20 °C	24.5 h VI (HL)	
		40 °C	1.7 h VI (HL)	
		60 °C	8.5 min VI (HL)	
		80 °C	0.59 min VI (HL)	

RNA: Viral genetic material detected by PCR. VI: Viable virus detected in cell culture. SBA: Viable virus detected by swine bioassay. HL: Half-life. NR: Not reported. ND: Not detected.

**Table 6 vetsci-11-00022-t006:** Survival of PRRSV in pig meat.

	Strain	Temp (°C)	Sample	Persistence	Reference
Pig muscle	MN-184	−20 °C and 4 °C	Ham region muscle	30 days at −20 °C, afterwards 7 days at 4 °C (RNA and SBA)	[50]
	Lelystad/SDSU #73	−23 °C and 4 °C		10 days at −23 °C, afterwards 1 day at 4 °C (RNA and VI)	[54]
Fresh pork sausage	VR-2332	4 °C	Pork sausage mix	15 days VI	[55]
		−20 °C		30 days VI	
Pig meat	North American strains (wild and vaccine-like)	−21 °C to −24 °C	Fresh pork	15 weeks RNA	[53]
	VR-2332 at 10^3^ TCID_50_/mL	4 °C		3 days VI	[56]
		25 °C		48 h VI	
		−20 °C		7 days VI	
	VR-2332 at 10^4^ and 10^5^ TCID_50_/mL	4 °C		6 days VI	
		25 °C		48 h VI	
		−20 °C		60 days VI	

RNA: Viral genetic material detected by PCR. VI: Viable virus detected in cell culture. SBA: Viable virus detected by swine bioassay. HL: Half-life. NR: Not reported. ND: Not detected.

## Data Availability

Data contained within this article.

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
