# Peer review of "Survival of Porcine Reproductive and Respiratory Syndrome Virus (PRRSV) in the Environment"

_vetsci, 2024, doi:10.3390/vetsci11010022_

Round 1
Reviewer 1 Report
Comments and Suggestions for Authors
While this review does not contain new significant scientific finding or interpretation, the authors provided an in-depth collection of PRRSV survival data. The review could be useful for different organisations trying to reduce the risk of PRRSV transmission. I recommend the manuscript.
Author Response
While this review does not contain new significant scientific finding or interpretation, the authors provided an in-depth collection of PRRSV survival data. The review could be useful for different organizations trying to reduce the risk of PRRSV transmission. I recommend the manuscript.
Answer: We thank the reviewer for this comment

Reviewer 2 Report
Comments and Suggestions for Authors
In this manuscript titled “Survival of Porcine Reproductive and Respiratory Syndrome Virus (PRRSV) in the Environment”, This article discusses the importance, transmission routes, and sources of infection of Porcine Reproductive and Respiratory Syndrome (PRRS), as well as the environmental stability and persistence of PRRS virus. The manuscript is well-organized and has certain significance. However, there are still a lot of problems in this manuscript that need to be revised, I would suggest accepting it after the following concerns are addressed.
1. The language needs considerable attention.
2. Please pay attention to writing format issues, such as proper capitalization and spacing. For example, in line 16, include is changed to include; in line 26, high is changed to a high; in line 162, virus was changed to viruses; add the before line 196 Mean; in line 239, PRRSV-positive is changed to PRRSV-positive; add the before line 302 live; in line 327, in a similar way suggested changing it to similarly; add a before line 389 low; add a before low in line 468.
3. The citation format appears to be unusual. In an article, the reference number should be placed within square brackets.
4. In the list of references, why are some documents marked with DOI, while others are not? Please unify the format of references.
5. The sentence lacks sufficient information in the context. It is recommended to increase the content.
6. The meaning of 'a ~' in line 39 is unclear. This expression is not standardized, and it is recommended to revise it.
7. The table formatting is disordered. It is recommended to reformat and edit the table.
8. The manuscript mentions experimental studies on the survival of PRRSV in various different environments, citing several research findings. However, in the discussion section, the author fails to reflect their own thoughts on how to better prevent and control PRRS in the future. While summarizing the findings, the author should incorporate their own insights and viewpoints.
9. As a review article, the referenced literature is outdated, and it is recommended to include more recent research findings. It is necessary to incorporate the latest research results into the article.
10. The word "naïve" in line 296 is misspelled.
11. The article mentions various research findings from different researchers, emphasizing the need to highlight their interconnectedness. It is important to incorporate personal insights and provide a comprehensive summary of the studies. Enhancing the coherence and logical flow among different sections of the article is crucial.
Author Response
- The language needs considerable attention:
Answer: We have revised the language extensively.
- Please pay attention to writing format issues, such as proper capitalization and spacing. For example, in line 16, include is changed to include; in line 26, high is changed to a high; in line 162, virus was changed to viruses; add the before line 196 Mean; in line 239, PRRSV-positive is changed to PRRSV-positive; add the before line 302 live; in line 327, in a similar way suggested changing it to similarly; add a before line 389 low; add a before low in line 468:
Answer: We thank the reviewer for this comment and have done the needful.
- The citation format appears to be unusual. In an article, the reference number should be placed within square brackets:
Answer: Done as suggested.
- In the list of references, why are some documents marked with DOI, while others are not? Please unify the format of references:
Answer: Done as suggested.
- The sentence lacks sufficient information in the context. It is recommended to increase the content: We are not sure which sentence is being referred to. However, we have extensively revised the language in the manuscript and hope that we have taken care of this commenting making the revision.
- The meaning of 'a ~' in line 39 is unclear. This expression is not standardized, and it is recommended to revise it:
Answer: Done as suggested.
- The table formatting is disordered. It is recommended to reformat and edit the table:
Answer: The format for the tables was taken from the journal’s template. However, following this comment, they were rechecked to make sure the formatting was ordered.
- The manuscript mentions experimental studies on the survival of PRRSV in various different environments, citing several research findings. However, in the discussion section, the author fails to reflect their own thoughts on how to better prevent and control PRRS in the future. While summarizing the findings, the author should incorporate their own insights and viewpoints:
Answer: This is a good comment and we have tried to improve the discussion in the revised manuscript.
- As a review article, the referenced literature is outdated, and it is recommended to include more recent research findings. It is necessary to incorporate the latest research results into the article:
Answer: We are unsure why the reviewer has made this comment. We have already cited papers from recent years.
- The word "naïve" in line 296 is misspelled:
Answer: Done as suggested.
- The article mentions various research findings from different researchers, emphasizing the need to highlight their interconnectedness. It is important to incorporate personal insights and provide a comprehensive summary of the studies. Enhancing the coherence and logical flow among different sections of the article is crucial.
Answer: This is a good comment and we have tried to improve the discussion in the revised manuscript.
